# The effect of age on the immediate recall of source vs. goal information from Czech sentences

Eva Pospíšilová

Faculty of Arts, Charles University

eva.pospisilova97@seznam.cz

**Background:** Earlier studies have observed systematic differences in speakers' ability to remember information from sentences [1]. Research focusing on the English language has shown that English speakers tend to recall the actor and subject of a sentence more easily, whereas information conveyed through verbs, adjectives, or adverbs is more difficult to remember [2]. These systematic differences have also been observed among Czech speakers; people tend to remember information conveyed by the object better than information provided by attributes or adverbial adjuncts specifying place or time [3, 4]. However, there is currently a gap in both Czech and international literature regarding research on sentence recall in relation to developmental aspects. While tasks focusing on sentence processing in adults have observed so-called good-enough processing mechanisms [5], reliance on good-enough processing has been less frequently shown in children and adolescents [6]. Therefore, it can be expected that the ability to memorize and recall information from a sentence will differ between children and adults and that these abilities will change during cognitive development. The present study thus investigates the effect of age on the ability to immediately recall information from sentences in children aged 11 to 17 and adults, focusing on the recall of information from adverbial adjuncts specifying direction/goal and source/place of origin. The selection of these specific types of adjuncts was based on findings from previous studies that pointed to a bias for goal information in language processing [7].

**Method:** Using the online dictionaries [Vallex 4.5](#) [8] and [SynSem Class Lexicon 5.0](#) [9], a total of 24 Czech transitive verbs were selected that can be combined with adverbial adjuncts conveying both information about the source and the direction/goal of the action. For each verb, a set of 4 sentences was created, two of which contain the verb in combination with an adverbial adjunct conveying information about direction, while the other two feature the verb with information about the source. The placement of these adverbial adjuncts also differed across the 4 sentences (see Table 1). In total, the experiment will contain 24 experimental and 72 filler sentences, which are currently being created. The experiment will use a self-paced reading method, each sentence will be immediately followed by an open-ended comprehension question targeting the information from adverbial adjunct conveying goal/source information. The type of sentence and question from each experimental item will be assigned to participants according to a Latin square design. The experiment will be tested on a group of children aged 11 to 17 years and on a control group of adults over 20 years of age.

**Expected results:** Consistent with previous studies, we expect to observe systematic differences in sentence recall among adult speakers, specifically that Czech adults will recall goal information better than source information. For the youngest group of child participants, we expect no such differences, as they likely have not yet acquired mechanisms of the good-enough language processing. Furthermore, we anticipate that the degree of systematic differences in recalling information from the two selected adverbial adjuncts will be modulated by age, with more systematic differences emerging as participants get older.

**Key words:** information recall, self-paced reading, source, goal, developmental aspects

## Tables

| type of advb. adjunct | position of adjunct | sentence |
|---|---|---|
| origin | beginning | Z chalupy maminka v sobotu přivezla opravdu tlustou starou knihu. |
| origin | middle | Maminka v sobotu přivezla z chalupy opravdu tlustou starou knihu. |
| direction | beginning | Na chalupu maminka v sobotu přivezla opravdu tlustou starou knihu. |
| direction | middle | Maminka v sobotu přivezla na chalupu opravdu tlustou starou knihu. |

Table 1: Experimental item example

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
