# OpenReview forum: "The effect of age on the immediate recall of source vs. goal information from Czech  sentences"
_CUNI.cz/2024/CJOLPhD — CUNI 2024 CJOLPhD Submission_

### Official Review · ~Maria_Onoeva1 · 2025-01-07
**Good job!**

The abstract outlines a planned self-paced reading study that will investigate how age affects the ability to immediately recall information from sentences. It’s well-written and structured, and the text is easy to follow. The research question is clear, but I have a few comments about the design of the experiment (see below). Including an example item really helped clarify the study design, but you’ll need to translate it if you’re planning to submit the abstract to an international conference. One more thing, I’m a bit confused by the terminology. Is the distinction between direction/goal and source/place of origin? In the example table and the text, you seem to use both, so maybe it would make sense to stick to one from the two.

1) What exactly do you mean by "an open-ended comprehension question"? Are participants supposed to type their answers? What is going to be measured as the dependent variable?
2) Why is this a self-paced reading study? Are you measuring reading time too? If so, do you have any predictions about age, reading speed and recall?
3) Since you’re manipulating the position of the adjuncts, do you have any predictions about how that might affect the results?

Good luck with the study!

---

### Official Review · ~Anna_Staňková1 · 2025-01-07
**Great!**

This is a well-written abstract about a planned study. It has a clear structure and itt is easy to follow. I like how you use numbers in brackets for citations - it saves space, yet you get the important references in the text. I also appreciate that you hyperlinked Vallex and SynSemClass Lexicon since it makes access a lot easier. In the method part, I would suggest describing the item as a 2 x 2 design (2 variables with two values), not as "a set of 4 sentences".

---

### Official Review · ~Radek_Šimík1 · 2025-01-08
**Very nice and clearly structured abstract!**

I have some contents-related comments and a few minor ones.

Comments on substance: It's a bit unclear to me whether there's little evidence for good-enough processing in children because so few people have tried so far, or because people have failed to find it; in other words, it's unclear whether [6] shows evidence for or against it. Normally it's fine to remain a bit ambivalent in these cases, but this is very central to your overall aims.

More out of curiosity: is good-enough processing something that has to be learned? Isn't it a very general mechanism that one starts with (and potentially needs to avoid it if necessary - when one needs to concentrate hard). I'd appreciate a bit more info or references on this issue.

- A suggestion for shortening: "Research focusing on the English language has that" could be dropped completely.
- Since the sentence that follows says basically the same for Czech, the two claims can be connected. E.g. "It has been shown that speakers... (for English: [2]; for Czech: [3, 4])."
- "in both Czech and international literature" could be dropped (generally, I'd advise you to avoid this opposition altogether, it's a bad habit of Czech linguists to speak about "Czech" vs. "international"... it's all just science)
- I'd add an "and" between "self-paced reading method" and "each..."
- "a Latin square design" - I guess "the Latin square design" makes more sense (it's a unique concept)